# Urolithin A Alleviates Doxorubicin-Induced Senescence in Mesenchymal Stem Cells

**DOI:** 10.3390/ijms262110257

**Published:** 2025-10-22

**Authors:** Alexander Kalinin, Ekaterina Zubkova, Mikhail Menshikov, Yelena Parfyonova

**Affiliations:** 1National Medical Research Centre of Cardiology Named After Academician E.I. Chazov, Moscow 121552, Russia; 2Faculty of Fundamental Medicine, Lomonosov Moscow State University, Moscow 119991, Russia

**Keywords:** cellular senescence, senotherapeutics, doxorubicin, aging, autophagy, mesenchymal stem cells

## Abstract

The accumulation of senescent cells, characterized by a pro-inflammatory secretory phenotype (SASP), metabolic dysfunction, and irreversible cell cycle arrest, is a driving force behind numerous age-related pathologies and directly undermines the therapeutic potential of mesenchymal stem cells (MSCs). In this study, we explore the senotherapeutic potential of urolithin A, a renowned antioxidant compound, in human adipose-derived MSCs (AD-hMSCs). Our findings reveal that urolithin A is non-cytotoxic to senescent AD-hMSCs and significantly suppresses the SASP by reducing the secretion of key pro-inflammatory mediators, including MCP1, PAI2, and IL1B. In addition, it was demonstrated that urolithin A was capable of reversing the decline in H3K9me3 levels induced by Doxorubicin treatment, restoring them to levels observed in untreated cells. The results of this study suggest that urolithin A functions as a senomorphic agent, capable of modulating cellular senescence. Moreover, its combination with senolytic therapies has the potential to yield novel and effective treatment strategies for regenerative medicine.

## 1. Introduction

Cellular senescence is a fundamental biological process that plays a dual role in determining cell fate. Transient senescence contributes to tissue regeneration and protects against tumor development [1,2].

The clearance of senescent cells relies on a fully functional immune system. When this system is compromised due to conditions like immunodeficiency or age-related decline, the immune cells themselves may undergo senescence (a process known as immunosenescence), diminishing their ability to eliminate dysfunctional cells [3]. This impaired clearance intensifies deleterious effects, fueling disease progression and exacerbating inflammation.

Currently, a plethora of compounds has been identified with anti-senescence properties, collectively referred to as senotherapeutics [4]. These include senolytics, which aim to safely and effectively eliminate senescent cells, and senomorphics, which specifically target the detrimental effects of the senescence-associated secretory phenotype (SASP) [4]. However, the absence of a specific and reliable marker of senescence has hindered the development of safe and effective senotherapeutic strategies. Furthermore, several compounds initially reported to possess anti-senescence activity have subsequently been shown to induce cellular senescence under different experimental conditions [5,6]. Addressing these limitations requires comprehensive research into known anti-senescent compounds, alongside the design of next-generation senotherapeutics with improved specificity and efficacy.

Urolithin A (3,8-dihydroxy-6H-benzo[c]chromen-6-one), a benzo-coumarin derivative, is a natural metabolite produced by gut microbiota through the conversion of dietary ellagitannins and ellagic acid, which are abundant in foods such as pomegranate, berries, and nuts [7]. Previously, it was demonstrated that urolithin A exerts protective effects against a range of pathological conditions, including neuroinflammation, myocardial dysfunction, cartilage degeneration, and acute kidney injury [7]. These beneficial effects are thought to be mediated, at least in part, by its ability to regulate mitophagy and mitochondrial function, thereby conferring protection against inflammation [8]. Furthermore, urolithin A has been demonstrated to mitigate replicative senescence in human skin fibroblasts as well as to attenuate oxidative stress-induced senescence in rat nucleus pulposus-derived mesenchymal stem cells and murine auditory cells [8,9,10]. Despite these promising findings, there remains a significant gap in our understanding of urolithin A’s effects across diverse human senescence models.

In this study, we researched the effects of urolithin A on a model of doxorubicin-induced senescence in human adipose-derived mesenchymal stem cells (AD-hMSCs), a cell population vital to tissue repair and regeneration. Given their pivotal role in maintaining tissue homeostasis and their therapeutic potential in regenerative medicine, preserving MSC functionality under senescence-inducing conditions is of critical importance. We propose that urolithin A may serve as an effective senescence-modulating compound with potential applications in enhancing MSC-based regenerative interventions

## 2. Results

### 2.1. Urolithin A Does Not Affect Proliferation of AD-hMSCs

To determine whether urolithin A protects AD-hMSCs from doxorubicin-induced damage or exerts senolytic activity, we assessed cell proliferation using the Cell Counting Kit-8 (CCK-8) assay. Doxorubicin treatment significantly reduced proliferation, indicating cell cycle arrest. Urolithin A, however, had no effect on proliferation in either control or doxorubicin-treated cells and did not induce cell death in either condition. AD-hMSC pretreatment with urolithin A also failed to prevent doxorubicin-induced cell cycle arrest (Figure 1A). Western blot analysis showed that post-treatment with urolithin A increased the expression of p21, a checkpoint inhibitor and senescence marker (Figure 1B,C). These findings suggest that urolithin A neither reverses senescence-associated cell cycle arrest nor exhibits senolytic activity in this model.

### 2.2. Urolithin A Does Not Alleviate Senescence-Associated Lysosomal Dysfunction

One of the hallmarks of cellular senescence is a deregulated metabolic profile, which encompasses mitochondrial and lysosomal dysfunction [11]. In this regard, senescent cells are commonly defined by the enhanced activity of senescence-associated beta-galactosidase (SA-βGal), an enzyme present in dysfunctional lysosomes with an altered pH.

X-gal staining of AD-hMSCs revealed minimal SA-βGal activity in healthy cells, whereas doxorubicin treatment markedly increased β-galactosidase activity. Treatment of senescent AD-hMSCs with urolithin A did not attenuate the doxorubicin-induced elevation in SA-βGal activity (Figure 2). These data indicate that urolithin A does not mitigate the doxorubicin-associated increase in lysosomal β-galactosidase activity under these conditions.

### 2.3. Urolithin A Suppresses the Secretory Phenotype of Senescent AD-hMSCs

Senescent cells, in addition to cell cycle arrest and dysregulated metabolism, are characterized by the acquisition of the senescence-associated secretory phenotype (SASP) formation [11]. SASP is commonly composed of various pro-inflammatory cytokines, chemokines, and matrix metalloproteinases. The detrimental effects of senescent cells are largely attributed to SASP, which sustains low-grade chronic inflammation and thereby promotes age-associated pathologies, including ‘inflammaging’ [12].

An emerging class of anti-senescence agents, termed «senomorphics», has been identified. In contrast to senolytics, which preferentially eliminate senescent cells by inducing cell death, senomorphics mitigate the deleterious effects of senescent cells, primarily by suppressing or reprogramming SASP [13].

Interleukin 1 beta (*IL1B*) and monocyte chemoattractant protein-1 (*MCP1*) are widely recognized as prominent components of the SASP and are included in the SenMayo gene set—a curated collection of 125 genes linked to the SASP [14]. This gene set has been validated in human bone biopsies and murine models, underscoring its value in detecting senescence and evaluating senescence-associated mechanisms. Additionally, plasminogen activator inhibitor 2 (*PAI2*) has been implicated in pro-inflammatory responses and the pathophysiology of cellular senescence [15,16].

To investigate whether urolithin A exhibits senomorphic activity, we performed RT-PCR and multiplex cytokine assay to evaluate its impact on key pro-senescent factors. First of all, we performed a brief RT-PCR analysis and demonstrated that urolithin A effectively reduced doxorubicin-mediated increases in pro-inflammatory factors *IL1B*, *MCP1*, and *PAI2* (Figure 3).

To further evaluate the potential senomorphic activity of urolithin A, we profiled conditioned media collected from AD-hMSCs using a multiplex cytokine assay. Urolithin A attenuated the doxorubicin-induced increases in eotaxin, FMS-like tyrosine kinase 3 ligand (FLT3L), interleukin-6 (IL6), tumor necrosis factor-α (TNF-α), tumor necrosis factor-β (TNF-β; lymphotoxin-α/LTA), and MCP-1 (Figure 4). IL6, TNF-α, TNF-β, and MCP1 are well-established SASP components and are included in the SenMayo gene set [14].

Consistent with their roles in pro-inflammatory signaling, eotaxin and FLT3L have also been implicated in pathophysiology of cellular senescence [17,18]. Together with the transcriptional data, these findings further support a senomorphic effect of urolithin A in doxorubicin-treated AD-hMSCs.

### 2.4. Urolithin A Restores Levels of H3K9me3 in Senescent AD-hMSCs

Cellular senescence is accompanied by extensive chromatin remodeling, including the formation of senescence-associated heterochromatin foci (SAHF) and changes in the epigenetic landscape [19]. Reduced levels of histone methylation marks, such as H3K4me3 and H3K9me3, are among the distinguishing features reported in senescent cells [20]. In our model, we observed that doxorubicin significantly decreased H3K9me3 levels. Notably, treatment of senescent in AD-hMSCs with urolithin A restored H3K9me3 to levels comparable to those in untreated cells (Figure 5). These findings suggest that urolithin A may exert its anti-senescent effects, at least in part, through modulation of chromatin state.

### 2.5. Doxorubicin-Induced Senescence in AD-hMSCs Is Characterized by Decreased Autophagy

Autophagy is a highly conservative fundamental cellular process that plays a dual role in the senescence pathophysiology. In particular, autophagy counteracts ER-stress and facilitates the production of SASP factors [21].

We evaluated autophagic status by Western blotting for sequestosome 1 (SQSTM1)—an adaptor protein that links ubiquitinated cargo to the autophagosome membrane—and by quantifying the ratio of microtubule-associated protein 1 light chain 3 isoforms (LC3-II/LC3-I), a commonly used marker of autophagosome maturation. Surprisingly, doxorubicin-treated AD-hMSCs exhibited changes consistent with reduced autophagic activity (Figure 6), namely, elevated SQSTM1 levels (Figure 6B) and a decreased LC3-II/LC3-I ratio (Figure 6C).

Given previous reports that urolithin A activates mitophagy, which, in turn, led to various beneficial effects [22], we next investigated whether it could restore autophagic activity in doxorubicin-treated senescent AD-hMSCs. However, urolithin A did not reverse the doxorubicin-induced impairment of autophagy, as neither SQSTM1 levels nor the LC3-II/LC3-I ratio were significantly altered by treatment (Figure 6).

These data suggest that urolithin A does not restore general autophagy in a model of doxorubicin-induced senescence.

## 3. Discussion

This study focuses on the effects of urolithin A in a post-treatment model of doxorubicin-induced senescence in AD-hMSCs, highlighting its potential as a senotherapeutic agent. Unlike most previous studies, which primarily investigate pre-treatment models where urolithin A is administered prior to exposure to a senescence inducer [8,9], our work explores whether urolithin A can affect cells in which senescence is already established.

Our findings demonstrate that urolithin A exerts a senomorphic, rather than senolytic, effect in the post-treatment model by reducing the secretion of pro-inflammatory SASP factors. This distinction is important: while pre-treatment models mainly assess urolithin A’s ability to prevent the senescence onset, post-treatment approaches are more relevant for evaluating whether a compound can alleviate the deleterious consequences of already-established senescent cells.

In our view, decreases in SA-β-Gal activity and p21 expression observed in post-treatment models more likely reflect senolytic activity—that is, the selective elimination of senescent cells—rather than genuine reversal of these markers within surviving cells. This interpretation aligns with our findings, showing that urolithin A does not exhibit senolytic activity in AD-hMSCs. At the same time, urolithin A does not reduce SA-βGal activity in doxorubicin-treated AD-hMSCs, yet it enhances the expression of p21, a key cell cycle inhibitor. Interestingly, this upregulation of p21 appears to be specific to senescent cells, as urolithin A had no impact on the proliferation of healthy AD-hMSCs.

Indeed, the context-specific regulation of p21 by urolithin A has been previously demonstrated: in pre-treatment senescence models, it has been shown to attenuate p21 expression [8,9], while in various cancer cell lines, it enhances p21 levels [23,24]. Taken together, these dual effects suggest that the effect of urolithin A on p21 is nuanced and context-specific, likely influenced by multiple factors. Moreover, increased p21 expression in senescent cells may not necessarily be detrimental. For instance, in in vivo models where p21-positive cells can be selectively eliminated, urolithin A mediated upregulation in p21 expression may play a positive role, since p21 protein is more profoundly associated with the deleterious senescence phenotype than other senescence-associated molecules, e.g., p16 protein [25]. Beyond its effects on p21, urolithin A showed a pronounced ability to suppress SASP, reducing the secretion of key pro-inflammatory factors such as IL1B, MCP1, PAI2, and others. This suppression aligns with the compound’s potential senomorphic activity, highlighting its capacity to mitigate the inflammatory burden of senescent cells without inducing their elimination. We also demonstrated that urolithin A modulates senescence-associated epigenetic landscape by restoring levels of H3K9me3 in senescent AD-hMSCs. We propose that this epigenetic effect may contribute to the downregulation of SASP factor expression. However, this potential mechanistic link requires rigorous validation through targeted perturbation of H3K9 methylation—using either genetic or pharmacological approaches—which will be the focus of future investigations.

In addition to that, several studies have investigated a potential link between the anti-senescence effect of urolithin A and its ability to stimulate mitophagy, a selective form of autophagy [8,22]. However, the role of autophagy—a fundamental cellular process—in the context of senescence remains a subject of ongoing debate [21]. A number of experimental studies have reported an activating effect of doxorubicin on autophagy [26,27]. Indeed, autophagy may contribute not only to the resistance to cellular stresses, but also to the synthesis of SASP factors [21]. Li and colleagues shed light on the function of autophagy in doxorubicin-mediated cellular damage, indicating that autophagy is indeed initiated following doxorubicin treatment, but over time, doxorubicin’s deleterious effects impair autophagy along with various other cellular processes, contributing to cellular dysfunction [28].

Consistent with this, our data indicate that autophagy is diminished in AD-hMSCs one week after doxorubicin-induced senescence. Interestingly, treatment with urolithin A does not restore autophagic activity. This suggests that, although urolithin A is recognized as a mitophagy activator, it may not broadly enhance general autophagy but rather selectively redirect autophagic processes toward mitochondrial turnover. While mitophagy was not directly assessed in this study, future investigations are warranted to clarify this mechanism and determine its potential contribution to the senomorphic effects of urolithin A.

It can be surmised that urolithin A may serve as a promising senomorphic agent capable of suppressing key features of the senescent phenotype in MSCs—most notably SASP and epigenetic dysregulation—without inducing senolysis. These properties support its potential utility in therapeutic strategies targeting age-related and inflammation-driven pathologies, particularly in the context of cell-based regenerative therapies. While our study has revealed only a fraction of this compound’s biological potential, further investigations are warranted to elucidate the underlying mechanisms and assess its translational applicability across diverse models of tissue degeneration and cellular senescence.

## 4. Materials and Methods

### 4.1. Cell Culture and Treatments

Human adipose-derived mesenchymal stem cells (AD-hMSCs) were obtained from the human biomaterials collection of the Institute of Regenerative Medicine (Lomonosov Moscow State University, collection ID: MSU_MSC_AD; repository catalog at https://human.depo.msu.ru/ (accessed on 8 August 2025). AD-hMSCs were cultured in DMEM-GlutaMAX^TM^ medium (Thermo Fisher Scientific, Waltham, MA, USA) supplemented with 10% fetal bovine serum (Cytiva, Marlborough, MA, USA) and 1x antibiotic-antimycotic solution (Biowest, Nuaillé, France) and cultivated under standard conditions (5% CO_2_, 37 °C). Cellular senescence was induced by treatment of AD-hMSCs with 500 nM doxorubicin (BLDpharm, Shanghai, China). Urolithin A (10 μM) was added for one day following a one-week period after the wash off of doxorubicin (post-treatment model).

### 4.2. Cell Viability Assay

To assess AD-hMSCs’ proliferation rate, we used cell counting kit 8 (CCK8; Dojindo, Kumamoto, Japan). AD-hMSCs were seeded with 4000 cells per well in a 96-well plate. In the case of the pre-treatment model, cells were treated with 10 μM urolithin A (BLDpharm, Shanghai, China) for 1 day. The next day, the senescence induction was performed by two days of treatment of the cells with doxorubicin. Cells were incubated for 1 week after washing off the senescence inducer. As for the post-treatment model, the cells were then treated with 10 μM urolithin A (BLDpharm, Shanghai, China) for 1 day. After that, referring to the manufacturer’s protocol, 10 μL/well CCK8 reagent was added to the cells. After incubation at 37 °C for 4 h. Optical density values in 96-well plates were measured at 450 nm in Victor™ X3 Multi Label Plate Reader (Perkin Elmer, Waltham, MA, USA). The 10 µM dose of urolithin A was selected following CCK-8 viability assays conducted across 10–100 µM, which confirmed it to be non-cytotoxic and suitable for downstream applications (Appendix A).

### 4.3. Senescence-Associated Beta-Galactosidase (SA-βGal) Staining

SA-βGal staining was performed as described by Debacq-Chainiaux et al. [29]. Briefly, AD-hMSCs were washed twice with PBS and fixed in formaldehyde-glutaraldehyde solution. Then the cells were double-washed and stained in the mix of pH buffer with X-gal overnight at 37 °C in a CO_2_-free incubator. After incubation, cells were washed twice with PBS and once with methanol, air-dried, and imaged by bright-field microscopy using an AxioObserver A2 inverted microscope (Carl Zeiss, Oberkochen, Germany). Five random images were taken for each donor within a single condition. SA-βGal staining intensity was assessed using ImageJ software version 1.54p.

### 4.4. Western Blot Analysis

AD-hMSCs were lysed in RIPA buffer with protease inhibitors (Protease Inhibitor Cocktail, HY-K0010, MedChemExpress, Junction, NJ, USA) and phosphatase inhibitors (Phosphatase Inhibitor Cocktail II, HY-K0022, MedChemExpress, Junction, NJ, USA). Samples were resolved by 15% SDS-PAGE and electrotransferred to a polyvinylidene difluoride membrane (Millipore, Burlington, MA, USA). After blocking with 5% non-fat milk in TBST for 1 h, the membrane was incubated overnight at 4 °C with primary antibodies: anti-betaActin (#AC-15, Abcam, Cambridge, MA, USA); anti-SQSTM1 (#A19700, Abclonal, Wuhan, China); anti-LC3B (#A19665, Abclonal, Wuhan, China); and anti-p21 (#A22460, Abclonal, Wuhan, China). Appropriate horseradish peroxidase-conjugated secondary antibodies were purchased from Invitrogen, USA. Proteins were visualized using SuperSignalTM West Pico PLUS Chemiluminescent Substrate (Thermo Scientific, Waltham, MA, USA) by enhanced chemiluminescence in Fusion FX gel-documentation system (Vilber-Lourmat, Collégien, France) and quantified in Image Lab 6.0.1 software (BioRad, Hercules, CA, USA). Western blot data were analyzed following the Degasperi’s protocol [30].

### 4.5. Real-Time Polymerase Chain Reaction (RT-qPCR)

Total RNA was extracted using a RNeasy Mini Kit (Qiagen, Germantown, MD, USA) according to the manufacturer’s protocol. RNA concentration was measured using a NanoDrop 2000 spectrophotometer (Thermo Fisher Scientific, Waltham, MA, USA). First-strand cDNA was synthesized from one microgram of total RNA using the High Capacity cDNA Reverse Transcription Kit (Thermo Fisher Scientific, Waltham, MA, USA) following the manufacturer’s protocol. RT-PCR was performed on a StepOnePlus™ Real-Time PCR System (Applied Biosystems, Waltham, MA, USA). The reaction mixture contained a SYBR Green PCR Master Mix (Evrogen, Moscow, Russia), primers (10 pmol of each dNTP), and cDNA (1–5 ng cDNA). The control mixture contained all components, with the exception of the matrix, replaced by de-ionized water. After the initial denaturation stage (95 °C, 10 min), 40 amplification cycles were performed for all primer pairs with annealing and elongation at 60 °C for 60 s. The primer sequences (Evrogen, Moscow, Russia) are shown in Table 1. mRNA levels between probes were normalized by the mRNA levels of beta-actin (ACTB) as a housekeeping gene, and the 2^−ΔΔCt^ method was used to calculate the relative expression levels of genes. RT-qPCR using primers for *TNFa*, *IL6* and *TNFb* genes was performed to validate multiplex immunoassay data (Appendix A).

### 4.6. Multiplex Immunoassay (Magpix)

To assess cytokine secretion, doxorubicin-treated AD-hMSCs were treated with urolithin A (10 µM; BLDpharm, Shanghai, China) for 24 h. Cells were then washed three times with culture medium, and conditioned media were collected 24 h post-treatment. Supernatants were clarified by centrifugation (400× *g*, 5 min, then 2000× *g*, 10 min) and stored at −80 °C until analysis. The cytokines quantification in the conditioned media was performed using the MILLIPLEX MAP 41 Human Cytokine/Chemokine Magnetic Bead Panel (HCYTMAG-60K-PX4141, Merck, Darmstadt, Germany) in strict adherence to the manufacturer’s guidelines. Initially, protein standards were serially diluted. Then, 25 µL aliquots of each standard dilution, control mixes (supplied by the manufacturer), and conditioned media samples were pipetted into a 96-well assay plate in duplicate. This was followed by the addition of fluorescent magnetic beads conjugated with cytokine-specific capture antibodies into each well. The plate was sealed and subjected to an overnight incubation at 4 °C with gentle shaking. Post-incubation, the plate was washed three times to remove unbound substances. Subsequently, a biotinylated detection antibody was added to each well and incubated for 1 h at room temperature. After a series of washes, streptavidin phycoerythrin was added, and the plate was further incubated for 30 min. Fluorescence detection and quantification were conducted using the MAGPIX^®^ System (Luminex-Merck, Darmstadt, Germany). The acquired data were analyzed utilizing xPONENT software 4.3.229.0, and cytokine concentrations in the conditioned media were extrapolated from the standard curves.

### 4.7. Immunofluorescent Staining of AD-hMSCs

AD-hMSCs were grown on glass coverslips and incubated overnight with urolithin A. Cells were then fixed with 4% formaldehyde in PBS for 10 min at room temperature (RT), followed by permeabilization in PBS containing 0.25% Triton X-100 for 10 min at RT. After permeabilization, samples were blocked with 10% donkey serum and 3% BSA in PBS for 1 h at RT to prevent nonspecific binding. Coverslips were then incubated with primary antibodies against H3K9me3 (Novus Biologicals, Centennial, CO, USA) or isotype-matched controls, diluted 1:100 in 1% BSA/PBS, overnight at 4 °C. After washing three times with PBS, samples were incubated for 1 h at RT with appropriate Alexa Fluor^®^-conjugated secondary antibodies donkey anti-mouse Alexa Fluor^®^ 594 (Thermo Fisher Scientific, USA) diluted 1:1000 in 1% BSA/PBS. Finally, nuclei were counterstained with DAPI (1 µg/mL, 10 min at RT), and coverslips were rinsed with PBS and mounted in Vectashield^®^ Antifade Mounting Medium (Vector Laboratories, Newark, CA, USA). Immunofluorescence images were acquired using a Stellaris 5 inverted confocal microscope (Leica Microsystems, Wetzlar, Germany). To quantify fluorescence signals, we used ImageJ software, and the corrected total cell fluorescence was calculated according to the protocol described by Gavet and Pines [31].

### 4.8. Statistical Analysis

All the graphs, calculations, and statistical analyses were performed using GraphPad Prism software version 10.3.1 for Microsoft Windows (GraphPad Software, San Diego, CA, USA). To test whether the collected numerical data are normally distributed, the Shapiro–Wilk normality test was applied. The comparison of means between different groups of numerical variables was performed using one-way ANOVA. Homogeneity of variances was tested using Brown–Forsythe and Bartlett’s tests, and in the case of unequal SDs, Brown–Forsythe and Welch ANOVA test was applied. All data were distributed normally, results are expressed as mean and standard deviation (SD) as dispersion characteristics, and *p* value less than 0.05 (*p* < 0.05) was considered as statistically significant.

## 5. Conclusions

In summary, our findings provide a comprehensive evaluation of the effects of urolithin A on doxorubicin-induced senescence in AD-hMSCs. Urolithin A did not affect cell proliferation or cell cycle progression and exhibited no senolytic activity, as indicated by unchanged cell viability and increased expression of p21, a key senescence marker. Furthermore, urolithin A did not alleviate lysosomal dysfunction, as SA-βGal activity remained elevated, nor did it restore autophagic function, with no significant changes in LC3-II/LC3-I ratio or SQSTM1/p62 levels. However, urolithin A exhibited a notable ability to restore H3K9me3 levels and attenuated the SASP, reducing the expression and secretion of key pro-inflammatory factors such as IL1B, MCP1, PAI2, and others. This suppression is consistent with the compound’s potential senomorphic activity, highlighting its capacity to mitigate the pro-inflammatory burden of senescent cells without inducing their elimination. These findings suggest that urolithin A may serve as a promising candidate for senomorphic therapy, particularly in contexts wherein modulating the SASP could alleviate inflammation-driven pathologies.

Further studies should delineate the mechanisms by which urolithin A modulates chromatin state and SASP programs and evaluate its therapeutic potential in regenerative settings and age-associated diseases.

## Figures and Tables

**Figure 1 ijms-26-10257-f001:**
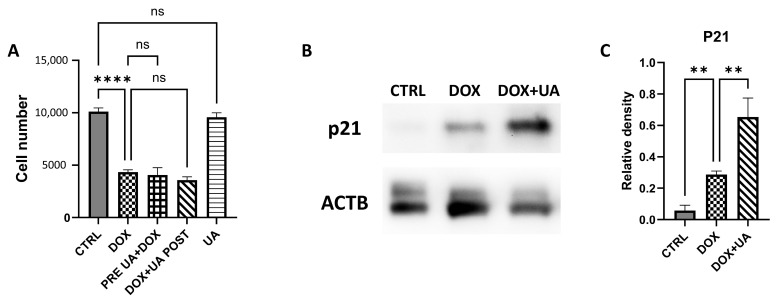
Urolithin A does not affect proliferation of healthy and senescent AD-hMSC. (**A**) Cell proliferation rate determined with CCK-8 assay. Data are expressed as a final cell number, where initial cell number per well was 4000. Doxorubicin («DOX»), Urolithin A («UA»); pre-treatment (PRE UA + DOX) group; post-treatment (DOX + UA POST) group. (**B**) Representative Western blot images showing cyclin-dependent kinase inhibitor 1A (p21; 21 kDa) and an internal control beta-Actin (ACTB; 45 kDa). (**C**) Graphical visualization of Western blot analysis of p21 protein (n = 3; «CTRL»—control group of intact AD-hMSCs; ** *p* ≤ 0.01; **** *p* ≤ 0.0001; «ns»—not significant).

**Figure 2 ijms-26-10257-f002:**
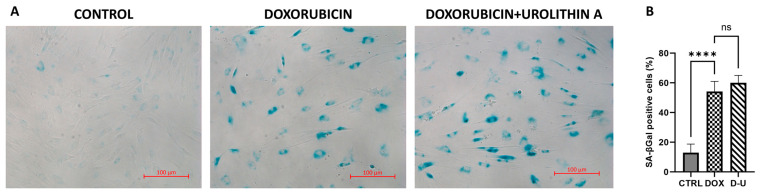
Senescence-associated β-galactosidase (SA-βGal) staining. (**A**) Representative images of cytochemical SA-βGal staining of AD-hMSCs («CONTROL»—a group of intact AD-hMSCs). (**B**) The quantification of SA-βGal activity in cytochemically stained cells. (n = 3; **** *p* ≤ 0.0001; «ns»—not significant).

**Figure 3 ijms-26-10257-f003:**
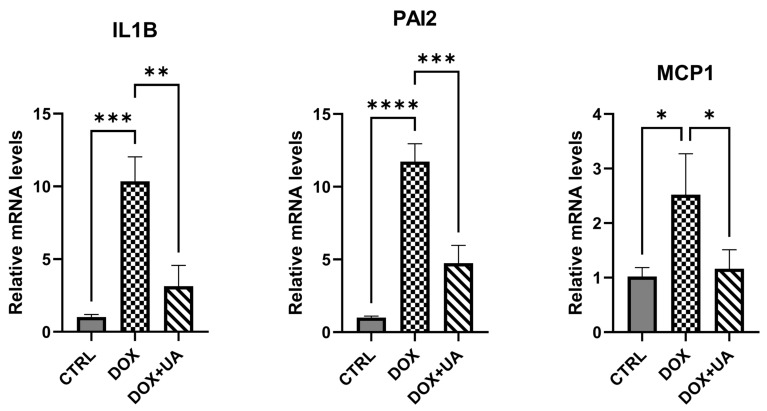
RT-qPCR analysis of SASP components alterations. RT-qPCR analysis was performed for each sample in duplicate. *IL1B*—interleukin 1 beta; *PAI2*—plasminogen activator inhibitor 2; *MCP1*—monocyte chemoattractant protein-1 (n = 3; **** *p* ≤ 0.0001; *** *p* ≤ 0.001; ** *p* ≤ 0.01; * *p* ≤ 0.05).

**Figure 4 ijms-26-10257-f004:**
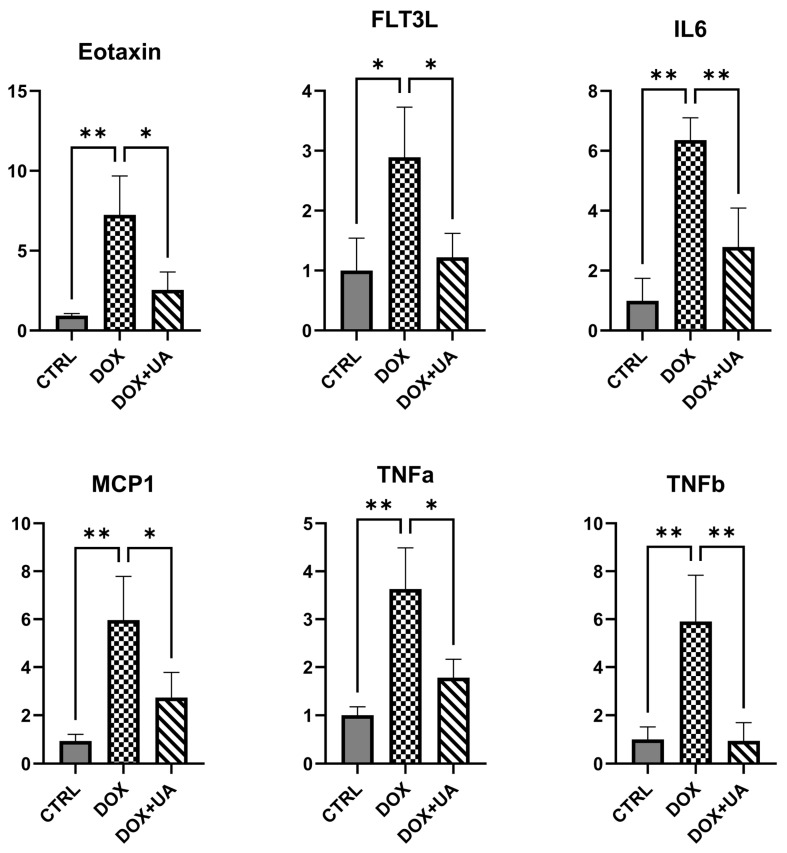
Multiplex cytokine profiling of conditioned media from AD-hMSCs by Luminex xMAP (MAGPIX). Cytokines analyzed were as follows: Eotaxin, FMS-like tyrosine kinase 3 ligand (FLT3L), interleukin 6 (IL6), tumor necrosis factor alpha (TNFa), tumor necrosis factor beta (TNFb), and monocyte chemotactic protein 1 (MCP1) (n = 3; ** *p* ≤ 0.01; * *p* ≤ 0.05). Data are shown as mean ± SD and expressed as fold change relative to control. Concentrations were quantified from standard curves per the manufacturer’s instructions.

**Figure 5 ijms-26-10257-f005:**
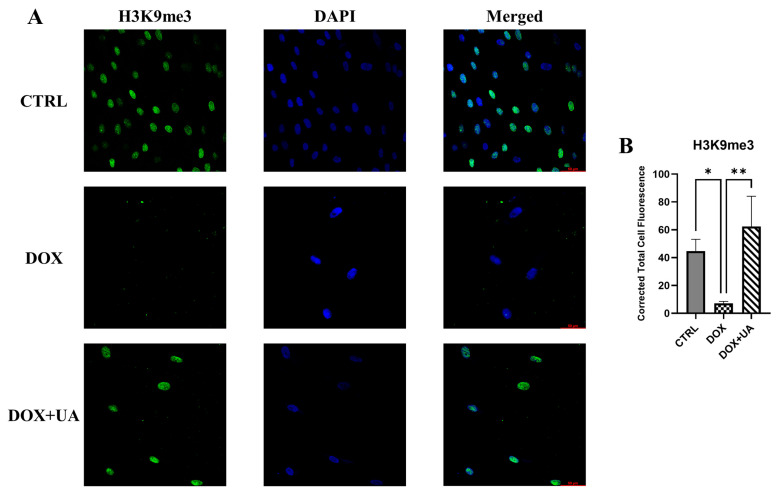
Fluorescent confocal microscopy images of H3K9me3 and DAPI. (**A**) Representative images of dramatic changes in H3K9me3 levels (DAPI staining was made to show nuclear localization of H3K9me3). (**B**) Quantitative analysis of H3K9me3 fluorescence intensity (n = 3; ** *p* ≤ 0.01; * *p* ≤ 0.05).

**Figure 6 ijms-26-10257-f006:**
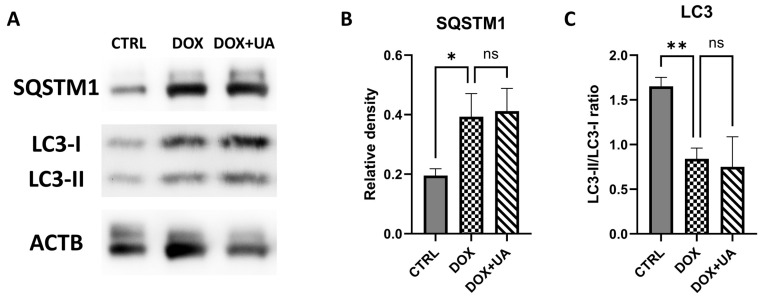
The assessment of DOX and DOX + UA combination effects on AD-hMSCs autophagy. (**A**) Representative photographs of Western blot of sequestosome 1 (SQSTM1; 62 kDa), light chain protein 3 forms 1 and 2 (LC3-I (16 kDa) and LC3-II (14 kDa), respectively), and a house-keeping protein beta-Actin (ACTB; 45 kDa). (**B**,**C**), relative expression of SQSTM1 (n = 3; * *p* ≤ 0.05) and LC3-II/LC3-I ratio (n = 3; ** *p* ≤ 0.01) are compared to the control values.

**Table 1 ijms-26-10257-t001:** List of PCR primers used.

Gene Name	Forward	Reverse
*ACTB*	CCTGGCACCCAGCACAAT	GGGCCGGACTCGTCATAC
*IL1B*	ACAGATGAAGTGCTCCTTCCA	GTCGGAGATTCGTAGCTGGAT
*MCP1*	CAGCCAGATGCAATCAATGCC	TGGAATCCTGAACCCACTTCT
*PAI2*	CATGGAGCATCTCGTCCAC	ACTGCATTGGCTCCCACTT

## Data Availability

The original contributions presented in this study are included in the article/Appendix A. Further inquiries can be directed to the corresponding author.

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
