# Peer review of "Urolithin A Alleviates Doxorubicin-Induced Senescence in Mesenchymal Stem Cells"

_ijms, 2025, doi:10.3390/ijms262110257_

Round 1

Reviewer 1 Report

Comments and Suggestions for Authors

General Comments

MSCs in different tissues are considered as a key actors to support  physiological homeostasis and regeneration. The present MS is focusing on the important issue of attenuation of already existing doxorubicin-induced  senescence of MSCs in vitro. The authors have implemented the experimental approaches well established in this area of research.  The conclusions are supported by the data presented.

Specific Comments

  1. It is necessary to use a certain style in abbreviations:

In Fig. 1, 3, Dox and D, Uro and U.  In Fig. 2, doxorubicin and Urolitin A without abbreviations, but Control and Ctrl in the same Figure. It’s confusing. In Figs 3, 5, please add an explanations of the abbreviations.

  1. As far as statistical analyses are concerned, the authors should avoid referring only to p less than 0.05, 0.001 etc. and try to give the exact p values to better report the levels of significance.

  1. Figure 2. Characterization of SA-βGal activity.

LL102-103: The percentage of SA-βGal positive cells as an indication of SA-βgal activity was quantified in n=5 images per condition.

In M&M. LL266-267: The dried dishes were photographed using bright field microscopy. SA-βGal staining intensity was assessed using ImageJ software.

Please, clarify this discrepancy.

“…n=5 images per condition…”. How were the view fields for these images chosen?

  1. It is better to use italic for gene symbols (IL1B etc) and IL-6, MCP-1 for proteins.

  1. Why does the list of analyzed genes differ from the SASP protein list? Would it be more informative to compare the transcription and secretion of the same proteins?

  1. Why the 500 nM doxorubicin and 10 μM Urolithin A concentrations were used?

  1. LL355-357 : “However, urolithin A exhibited a notable ability to restore levels of H3K9me3 and to suppress the senescence-associated secretory phenotype (SASP), reducing the secretion of key pro-inflammatory factors such as IL1B, MCP1, PAI2, and others. “

In MS, the data on MCP-1 protein level only is presented. Please, clarify.

Comments on the Quality of English Language

The Professional English editing is recommended.

Author Response

Dear Reviewer,

We would like to express our gratitude for your thorough review of our article. Your insightful comments have contributed to enhancing the readability and appeal of the article for readers of IJMS. We will address your comments in a point-by-point manner regarding the changes that have been made.

  1. We fully support your comment and have undertaken a comprehensive revision of the full text and all figures to ensure the designations of the study groups are now consistent throughout. The control group of intact AD-hMSC is designated as "CTRL"; AD-hMSC treated with doxorubicin are designated as "DOX"; and AD-hMSC treated with doxorubicin and urolithin A are designated as "DOX+UA." We believe that such uniform labeling in the figures will improve reader’s comfort, and we thank you for your valuable comment.

  1. We understand your concern, but we should note that introducing the value p for each difference directly in the figures themselves would make them overcluttered. Another option would be to use a different symbol for each significant difference, such as @ or #, but then, since the p-value varies throughout the article, a large number of similar symbols would have to be added, which would reduce the reader's comfort. The asterisk symbol for the p-value is easily recognizable and consistent throughout the article. Nevertheless, we thank you for your important comment and will consider possible ways to correct this shortcoming in the future.

  1. We extend our gratitude for this insightful commentary. To elucidate the methodological framework of our experiment, the following sentence was incorporated into paragraph "4.3. Senescence-Associated Beta-Galactosidase (SA-βGal) Staining" (Materials and Methods): "Five random images were taken for each donor within a single condition." Additionally, the sentence "The percentage of SA-βGal positive cells as an indication of SA-βgal activity was quantified in n=5 images per condition." has been removed from the description of Figure 2 and "n=3" has been added, which now reflects the number of donors rather than the number of photographs per donor-condition. We believe that this will improve the reader's experience of our article and make its methodology more understandable and transparent.

  1. We have taken your comment into account and indicated the genes in italics. For example, in the paragraph "Urolithin A suppresses secretory phenotype of senescent AD-hMSCs" - "Transcriptome analysis showed that urolithin A effectively reduced doxorubicin-mediated increases in pro-inflammatory factors interleukin 1 beta (IL1B), plasminogen activator inhibitor 2 (PAI2) and monocyte chemoattractant protein-1 (MCP1) (Fig. 3)." (Results) and in the table in the paragraph "Real-Time Polymerase Chain Reaction (RT-PCR)" (Materials and Methods)

  1. Thank you for this important comment. We realize that it was an oversight on our part not to fully disclose the logic of the experiment's framework. We used RT-PCR in the initial stage of testing urolithin A for a senomorphic effect. We selected some of the most upregulated factors in cellular senescence: IL1B, MCP1, and PAI2 according to the SenMayo database. Next, we performed a secretome analysis to validate the senomorphic effect of urolithin A. We demonstrated that urolithin A suppresses not only the expression of the selected genes, but also a broad spectrum of pro-inflammatory factors. We revised the section titled "Urolithin A Suppresses Secretory Phenotype of Senescent AD-hMSCs" (Results). Nevertheless, we have performed additional RT-qPCR to assess the expression of the SASP factors from MAGPIX assay and find out that the levels of the selected molecules were changed insignificantly, but with the same trend as in MAGPIX assay. This observation highlights the complexity of urolithin’s A senomorphic effect. We thank you for this critical advice and attach “Supplement 2” with results of the abovementioned RT-qPCR analysis. 

  1. We appreciate your valuable comment and have prepared supplementary material 1 containing detailed information for your review. To summarise, we used a cell viability assay to determine the sublethal dose of doxorubicin and the maximum non-toxic dose of urolithin A, and we are confident that this supplement will enhance the credibility of the article and improve the reader experience.

  1. We took your comment into account and replaced the word "secretion" in the indicated sentence with "expression" to avoid any misunderstanding on the part of readers. The text containing this improvement more accurately reflects the results of our experiments.

A.K., E.Z. and M.M. triple checked the English text for any inaccuracies and have made many improvements (which are visible in the text) in it.

Reviewer 2 Report

Comments and Suggestions for Authors

This study focuses on the intervention effect of urolithin A on the aging of human adipose derived mesenchymal stem cells (AD hMSCs) induced by doxorubicin. Through multidimensional experiments, the "aging morphology regulator" attribute of urolithin A is clarified, filling the research gap in its intervention mechanism in established aging models. The research design is logically coherent and the methodology is generally standardized, providing a new perspective for the functional repair of aging stem cells in regenerative medicine. However, there are still some issues as following.

  1. Clarify the background of cell donors, culture generations, and screening criteria for urolithin A concentration, and improve the validation criteria for establishing aging models.
  2. Add a mechanism validation experiment for "epigenetic regulation of SASP" (such as H3K9me3 inhibitor intervention), or explicitly propose this direction as a future research plan in the discussion to compensate for the insufficient depth of the mechanism.
  3. Lack of evaluation of urolithin A for the functional repair of AD hMSCs: only detecting changes in cell proliferation, aging markers, and molecular levels, without detecting core stem cell functions (such as adipogenic/osteogenic differentiation ability, paracrine factor secretion profiles other than SASP), unable to clarify its actual value in "improving the therapeutic potential of aging stem cells" (such as the absence of osteogenic differentiation Alizarin Red staining and adipogenic differentiation Oil Red O staining experiments).
  4. The selection criteria for the concentration of urolithin A are missing: Sections 4.1 and 4.2 both use 10 μ M urolithin A intervention, but do not specify the basis for determining this concentration (such as the screening results of pre experimental concentration gradients and the rationality analysis of reference concentrations in literature), which cannot rule out "false negatives/false positives caused by high/low concentrations"
  5. In disccusion, the speculation about “We suggest that this ability of urolithin A may underlie its suppressive effect on the SASP components expression”lack experimental verification, and the mechanism explanation is weak.
  6. The "Abbreviations" section has serious duplication: abbreviations such as AD hMSCs and CCK8 are listed three times, which does not meet the requirements of the journal format and wastes space.

Author Response

Response to Reviewer

We thank the reviewer for the thoughtful and constructive comments. We have revised the manuscript accordingly and address each point below.

1) Clarify the background of cell donors, culture generations, and screening criteria for urolithin A concentration, and improve the validation criteria for establishing aging models.

Response. Thank you for the careful review. We expanded the Materials and Methods to report donor tissue source and culture conditions.

We also added CCK-8 cell viability data in the Supplementary Materials, demonstrating that 10 µM urolithin A maintains viability comparable to vehicle controls, thereby justifying its use in our experiments.

In addition, we clarified the treatment conditions for doxorubicin-induced senescence (dose, exposure, washout, and recovery) and explicitly listed the validation criteria used to confirm senescence: SA-β-gal activity, p21 expression, alongside CCK-8 viability measurements.

2) Add a mechanism validation experiment for "epigenetic regulation of SASP" (such as H3K9me3 inhibitor intervention), or explicitly propose this direction as a future research plan in the discussion to compensate for the insufficient depth of the mechanism.

Response. We agree that causal linkage between H3K9me3 and SASP modulation requires targeted perturbation. While our current dataset shows that urolithin A restores global H3K9me3 and reduces SASP, we have intentionally tempered our interpretation and now explicitly recognize the epigenetic regulation of SASP as an important objective for future investigation.

In order to directly address this question, we have modified the “Discussion” section:

“We also demonstrated, that urolithin A modulates senescence-associated epigenetic landscape by restoring levels of H3K9me3 in senescent AD-hMSCs.  We propose that this epigenetic effect may contribute to the downregulation of SASP factors expression. However, this potential mechanistic link requires rigorous validation through targeted perturbation of H3K9 methylation—using either genetic or pharmacological approaches—which will be the focus of future investigations.”

3) Lack of evaluation of urolithin A for the functional repair of AD hMSCs: only detecting changes in cell proliferation, aging markers, and molecular levels, without detecting core stem cell functions (such as adipogenic/osteogenic differentiation ability, paracrine factor secretion profiles other than SASP), unable to clarify its actual value in "improving the therapeutic potential of aging stem cells" (such as the absence of osteogenic differentiation Alizarin Red staining and adipogenic differentiation Oil Red O staining experiments).

Response. We appreciate this valuable observation. In the current study, our primary focus was on characterizing senescence markers and SASP modulation. In response to this point, we have carefully revised the manuscript to moderate statements related to the compound’s therapeutic potential and to ensure that our interpretations remain appropriately cautious.

4) The selection criteria for the concentration of urolithin A are missing: Sections 4.1 and 4.2 both use 10 μ M urolithin A intervention, but do not specify the basis for determining this concentration (such as the screening results of pre-experimental concentration gradients and the rationality analysis of reference concentrations in literature), which cannot rule out "false negatives/false positives caused by high/low concentrations»

Response. Thank you for this helpful comment. We have added CCK-8 cell viability data to the Supplementary Materials to support the selection of the 10 µM dose of urolithin A. Additionally, we have included a brief explanation in Sections 4.1 and 4.2 of the Materials and Methods to clarify the rationale for the selected concentration and to specify the assay conditions:

“The 10 µM dose of urolithin A was selected following CCK-8 viability assays conducted across 10–100 µM, which confirmed it to be non-cytotoxic and suitable for downstream applications (Supplementary materials).”

5) In discussion, the speculation about We suggest that this ability of urolithin A may underlie its suppressive effect on the SASP components expression” lack experimental verification, and the mechanism explanation is weak.

Response.

We appreciate the reviewer’s insightful comment. In response, we have revised the Discussion to moderate the speculative language regarding the potential mechanistic link between urolithin A–induced restoration of H3K9me3 and SASP suppression. Specifically, we now clearly state that this association remains hypothetical and requires rigorous experimental validation.

6) The "Abbreviations" section has serious duplication: abbreviations such as AD hMSCs and CCK8 are listed three times, which does not meet the requirements of the journal format and wastes space.

Response. Thank you for bringing this to our attention. The duplication in the Abbreviations section was the result of a formatting error during manuscript preparation. We have now corrected this and provide a single, non-redundant list in accordance with the journal’s guidelines.

Reviewer 3 Report

Comments and Suggestions for Authors

In this article, Kalinin et al, examine the roles of UA in doxorubicin-induced mesenchymal stem cells with senescence feature. Through this cellular senescence model, the author demonstrated that UA have on impact on the proliferation and lacked senolytic activity and did not alleviate lysosomal function and autophagy. However, UA specifically restore the levels of H3K9me3 and suppress the SASP with reducing the secretion of key pro-inflammatory factors. The author concluded that UA exerts a senomorphic, rather than senolytic, effect in the post-treatment model by suppressing the secretion of SASP factors. Overall, this study provided valuable insights for the function of UA in regulating cellular senescence. UA has studied as a popular anti-aging compound in recent years.

This manuscript provides valuable insights and encourages researchers to perform systematic evaluations of the anti-aging effects of UA across different models and disease contexts. The study is logically organized. However, I have several comments and suggestions aimed at improving the clarity and professional presentation of the manuscript before it can be considered suitable for publication.

Main comments,

1. Although the authors attempted to use abbreviations to distinguish different experimental groups in the figure, some of the abbreviations are overly brief, which makes it difficult for readers to quickly and clearly interpret the information. I recommend using more descriptive labels to enhance readability.

In Figure 1A, I suggest revising the labels in both the figure and the legend as follows: change “D-U PRE” to “PRE UA+DOX,” “D-U POST” to “DOX+POST UA,” and “URO” to “UA.” Similarly, in Figure 1B, “D-U” could be changed to “D+U.” Please ensure that the labeling across all figures is consistent and clearly presented.

2.⁠ ⁠ In Figures 1B and 6A, I suggest changing “bACT” to the more commonly used abbreviation “ACTB” for consistency with standard gene nomenclature.

In addition, in Figures 1B and 6A, please include the molecular weight information, either by indicating the expected size of the target proteins or by adding the protein ladder. This will make the western blot data more easier to interpret. 

3.⁠ In the quantitative analyses of the Western blot data shown in Figures 1C, 6B, and 6C, please present the fold changes relative to the CTRL group by setting the mean value of the CTRL group to 1.0. This adjustment will improve data clarity and facilitate comparison across experimental groups.

4.⁠ ⁠The authors appear to have used the same ACTB bands in Figures 1 and 6. Please clarify the source of these data.

5. Please include scale bars in Figures 2A and 5A.

6.⁠ ⁠Please specify the data presentation format (e.g., mean ± SD or mean ± SEM) and the statistical methods used in each figure legend to ensure clarity and reproducibility of the results.

7.⁠ In Figure 3, the authors used RT-qPCR to analyze few specific genes rather than RNA-seq to examine the entire transcriptome. Accordingly, I recommend changing the term “transcriptome” to “RT-qPCR” on Page 4, lines 121 and 132. Similarly, please consider using a more appropriate term to replace “secretome” that accurately reflects the experimental approach.

8. Please ensure that all vertical axes in Figure 4 are labeled to clearly indicate the measured parameters, which will improve the interpretability of the figure.

9.⁠ Since “CTCF” is a commonly used abbreviation for the “CCCTC-binding factor (CTCF) transcription factor,” please label the vertical axis in Figure 5B directly as “Corrected Total Cell Fluorescence” to allow readers to immediately and clearly understand the data.

Minor comments,

1.⁠ ⁠On page 3, lines 86 and 101, please change the word “intact AD-hMSCs” into “wild type AD-hMSCs with vehicle”.

2.⁠ ⁠On page 6, line 173, 174 and 178, It should be “Fig.6”.

3.⁠ ⁠On page 10, line 321, It should be “AD-hMSCs” instead of “ADSC”.

4.⁠ ⁠It would be better if the drug treatment time and concentration were described in each figure legend.

Author Response

Dear Reviewer,

We would like to express our gratitude for your thorough review of our article. Your insightful comments have contributed to enhancing the readability and appeal of the article for readers of IJMS. We will address your comments in a point-by-point manner regarding the changes that have been made.

  1. We fully support your comment and have undertaken a comprehensive revision of all figures to ensure the designations of the study groups are now consistent throughout. The control group of AD-hMSC is designated as "CTRL"; AD-hMSC treated with doxorubicin are designated as "DOX"; and AD-hMSC treated with doxorubicin and urolithin A are designated as "DOX+UA." We also change description of Figure 1A to "PRE UA+DOX", "DOX+POST UA", and "UA" for pretreatment, posttreatment and sole urolithin a treatment models, respectively. We believe that such uniform labeling in the figures will improve reader’s comfort, and we thank you for your valuable comment.

  1. In the article, we indeed used two abbreviations for the beta-actin protein: "bACT" and "ACTB." We concur with your assertion that the original designation for this protein is "ACTB" and that a uniform designation for beta-actin is required throughout the article. Consequently, all instances of "bACT" have been substituted with "ACTB," both in the text and in the images. We are appreciative of the observation that such a nuanced yet pivotal detail was brought to our attention.

We acknowledge your concern regarding the necessity of providing the molecular weight of the proteins. However, we must emphasize that incorporating this information would introduce unnecessary complexity to the figures, which could potentially confuse our readers. However, we concur with your assertion regarding the necessity of including the molecular weight of proteins. Consequently, we have added this information in the description of the figures.

  1. We acknowledge your concerns regarding the optimal representation of Western blot analysis results. In this case, we adhered to a specific protocol, as we now outline in the "Materials and Methods" section, paragraph "Western Blot Analysis". The relevant paragraph states: "Western blot data were analyzed following the Degasperi’s protocol [15]." We assume that excessive deviation from the protocol could mislead the reader. We appreciate your important observation.

  1. The observation that the ACTB bands coincide in Figures 1 and 6 is indeed valid. This occurrence is attributed to the fact that the proteins (p21, SQSTM1, and LC3II/I) were imaged on a single membrane. Furthermore, this evidence substantiates the notion that the alterations in the expression of the senescence-associated protein p21 are concomitant with the observed changes in the autophagy proteins SQSTM1 and LC3II/I. This demonstrates that we are not doing cherry picking, but rather present original, genuine data. Nevertheless, we thank you for the valuable comment.

  1. We appreciate your valuable advice and have added scale bars to the figures you mentioned. This is a very important comment, and we believe that this correction will definitely improve the reader's experience.

  1. We appreciate the commentary and have revised the manuscript. Specifically, the "Statistical Analysis" section states that “All data were distributed normally, results are expressed as mean and standard deviation (SD) as dispersion characteristic, and p value less than 0.05 (p < 0.05) was considered as statistically significant.” We deliberately chose not to include this information in the captions for each figure so as not to overload them. Thank you for this valuable observation. 

  1. We would like to express our gratitude for this valuable commentary. In order to prevent any potential misunderstandings on the part of readers, the terms "transcriptome" and "secretome" have been replaced with “RT-qPCR” and “multiplex cytokine assay”, respectively. We suggest that this improvement will enhance the reader's experience of our article and make its methodology more understandable and transparent.

  1. We appreciate your comment regarding the need to label the ordinate axis on the graphs. However, we decided to exclude them from the figure, leaving this designation in the captions: "Data are shown as mean ± SD and expressed as fold change relative to control. Concentrations were quantified from standard curves per the manufacturer’s instructions." This approach allowed us to circumvent the issue of overloading the figure and enhance the reader's experience.

  1. We fully support your comment and have decided to avoid the use of unnecessary abbreviations by replacing “CTCF” with “Corrected Total Cell Fluorescence” in Figure 5B. We have also removed all references to “CTCF” from the entire article to improve readability and to prevent potential misunderstanding on the part of the reader.

We have taken your minor comments into account and made the appropriate changes. However, we are unable to replace "intact AD-hMSCs" with "wild type AD-hMSCs with vehicle" because we used human cells in our experiment. Thank you again for your thoughtful and thorough review of our article. Your review helped us identify many omissions and significantly improve the article's readability.

Round 2

Reviewer 2 Report

Comments and Suggestions for Authors

There is no comment anymore.